# The Impact of Public Grants on Firm-Level Productivity: Findings from the Czech Food Industry

**Ondřej Dvouletý** [1],* and **Ivana Blažková** [2]

1   Department of Entrepreneurship, University of Economics, W. Churchill Sq. 1938/4, 130 67 Prague 3, Czech Republic
2   Department of Regional and Business Economics, Mendel University in Brno, Zemědělská 1, 613 00 Brno, Czech Republic; blazkova@mendelu.cz
*   Correspondence: ondrej.dvoulety@vse.cz; Tel.: +420-224098-753

**Abstract:** Studying the effects of public entrepreneurship and small- and medium-sized enterprise (SME) policies on productivity (i.e., technological efficiency) is important, because the investment policies primarily aim to reduce allocation inefficiencies, enable usage of economies of scale, promote new production methods and technological development. We reviewed the recently published studies, and we show that they often lack fundamental information, such as a sample description and numbers of supported and non-supported firms. Keeping in mind the importance of transparent and rigorous empirical evaluations, we evaluated the effects of investment support from the European Regional and Development Fund (ERDF) on the productivity of the firms operating in the Czech food processing industry two years after the end of the programme. Methodologically, we apply the propensity score matching approach (PSM) combined with a difference in differences approach (DID) based on the firm-level data accounting for 157 firms (i.e., 77.3% of all beneficiaries within the industry) and a control sample of 1224 firms that have not been supported by the intervention. We use three measures of productivity—production efficiency, labour productivity and total factor productivity (TFP). The obtained findings showed that investment subsidy had a positive impact on labour productivity of supported firms. However, the effects on TFP were negative. The impact on production efficiency indicator was proven to be inconclusive. It follows from the results that the productivity of subsidised firms did not improve through an internal increase in efficiency (efficiency of the use of inputs), which indicates no significant technological change. The subsidy decision-making processes should be more careful and transparent to ensure allocating resources only to the projects with growth potential.

**Keywords:** investment subsidy; capital grants; firm productivity; public policy evaluation; entrepreneurship and SME policy; counterfactual impact analysis; food processing industry; Czech Republic

## 1. Introduction

It has been widely acknowledged by the literature that stable economic growth is closely related to the entrepreneurial capacity of an economy [1–5], although there are studies showing that this relationship might differ across regions and continents [6–8]. Concerning the role of entrepreneurship in stimulating economic growth, the importance of carrying out innovations and innovative behaviour is emphasised [1,9–11]. Successful implementation of innovation brings significant growth in technical and economic efficiency of an enterprise reflected in its productivity [4,12–14], which brings important implications for sustainable long-term growth. This is widely addressed by public policies, the important goal of which is promoting innovation and technological progress to enhance competitiveness and

economic growth [15–19]. Given that productivity is considered as one of the most reliable indicators for competitiveness over the long term [20], the question arises whether the support leads to improvements of productivity and thus to fulfilling its purpose. Productivity indicators might thus answer the question, whether the public interventions succeed in boosting sustainable growth of the supported companies in the long term.

Despite the growing body of literature investigating the investment support policy contribution to competitiveness and efficiency of subsidised firms, e.g., references [21–30], the minority of studies focuses specifically on productivity effects, e.g., references [31–35]. The productivity assessment using only the labour productivity indicator, as common in a number of studies, e.g., references [25,36–40], may bias the overall productivity effects. Bernini et al. [32] argue that labour productivity may increase because of the capital deepening induced by the subsidy, whereas the efficiency of all inputs used may not increase at all. Likewise, production efficiency as an indicator of effective material utilisation evaluates productivity from one angle only—from the viewpoint of efficient material utilisation [41,42]. On that account, the total factor productivity (TFP) appears to be the most appropriate measure of productivity to analyse the effectiveness of a subsidised firm [31–33,42–45], because it accounts for the possible substitution in the factor usage within the production process, reflects the changes of total output that are not explained with increase of capital and labour, and allows the measurement of technological change [46–48]. Therefore, this study responds to this deficiency of existing research by involving all these productivity indicators in one analysis.

Although governments and policymakers assume that investment subsidies are growth enhancing, due to the acceleration of technological development and better utilisation of economies of scale [48], the empirical findings are not convincing. Despite the extensive number of studies on this issue, e.g., references [31,43,44,47,49], there is little consensus among economists on the effectiveness of investment incentives regarding productivity. Previous empirical findings and theoretical studies show that the effects of subsidies on productivity level may be both positive and negative. Whereas Bernini et al. [32], Howell [45] and many others document negative impact of subsidies on productivity as a result of allocative (and technical) efficiency, soft budget constraints and the shift of subsidies to less productive enterprises, there are also arguments for this support, e.g., references [31,40,43], particularly because of investment-induced productivity gains caused by the interaction of credit and risk attitudes with subsidies.

Our study strives to contribute to the debate on the productivity effects of public subsidies through the counterfactual analysis of the effects of the EU public policy, exemplified by the Operational Programme Enterprise and Innovation (OPEI) for the Czech Republic, which was targeted on the increase of competitiveness and innovation performance of the Czech industry [50]. The OPEI was implemented in the Czech Republic to draw support from the European Regional Development Fund (ERDF) in the period 2007–2013. The focus of the OPEI was to support especially small- and medium-sized enterprises (SMEs) and boost the entrepreneurial activity and economic growth through the investment subsidies [51]. It is also worth mentioning that the OPEI was the third largest Czech operational programme and it was the largest systematic promotion of entrepreneurship in the country by that time [50]. Our study aims to assess the effects of this support programme on the productivity of beneficiaries within the Czech food processing industry. Methodologically, we apply the propensity score matching approach (PSM) combined with a difference in differences approach (DID) based on the firm-level data accounting for 77.3% of all beneficiaries within the industry. In this study we build on prior research on the effects of EU investment support in the Czech agribusiness [39,40,52,53], which indicates overall positive impacts of investment subsidies on the financial performance of the supported enterprises. In contrast to previous studies that have in view only labour productivity as one of the performance indicators, we solely focus on productivity effects by employing more productivity indicators in one analysis, namely the production efficiency indicator, labour productivity and TFP, which increases the robustness of our findings.

Empirical research on the effects of investment support on productivity is important for more reasons. The results may contribute to identification of public policies and actions leading to the improvement of the competitive position of companies through the productivity-enhancing innovations [54,55], which brings higher level and growth rate of productivity in the long run, referred to as "dynamic efficiency" gains [56]. This is particularly important for the food processing industry, where there is a significant need for innovations [57] in order to satisfy the growing consumer demand for product quality, variety and convenience, to ensure food safety as well as to maintain efficient production, exploit economies of scale and meet environmental regulations [58]. Additionally, the food industry, as an interconnecting link between the primary agricultural production and the final consumer, brings to the marketplace the benefits of successfully implemented innovations through the spillover effects on the downstream and upstream markets, and thus significantly influences the competitiveness of the whole agribusiness sector [59,60] and continuous economic growth [61]. The Czech context is relevant with regard to the Czech food market development during the last decades, which has brought the need to promote technological progress that would boost the productivity growth (for details see reference [62]). Moreover, it is worth mentioning that the share of the food processing industry on the value added to the whole Czech manufacturing industry was 7.5%, and on employment, it was 9.2% in 2016 according to the Czech Statistical Office (cf. Blažková and Dvouletý [39]). As technological innovation and upgrading are regarded as key aspects of catch-up for transitioning countries [45], the support policy should encourage innovation and technological upgrading, and spur regional development.

In general, substantial financial resources are allocated by the European Union to the promotion of entrepreneurship in the belief that such public support is growth enhancing. But is that really so? Does allocation of public funds through entrepreneurship and SME policies lead to productivity gains and increase in competitiveness of the subsidised companies? It is still difficult to draw a cohesive conclusion. Therefore, rigorous impact evaluations of such policies need to be implemented to help policymakers with designing of the interventions that have a real impact on the target group. In addition, given the pressure on the effectiveness of EU funds, this is evidently a hot issue and thus we believe that our study has important empirical and methodological implications for both the evaluation community and policymakers.

The next parts of the paper proceed as follows. Theoretical background in the context of economic theory and previously published studies is given in Section 2. Section 3 describes data, variables and empirical approach. Section 4 presents the results of the counterfactual impact analysis, and the last part of the paper (Section 5) discusses the findings and policy implications and provides suggestions for future research.

## 2. Theoretical Background

Economic theory indicates a both positive and negative relationship between investment subsidies and productivity, e.g., references [44,63–65]. The positive relationship is usually anticipated by risk behaviour and limited funds, e.g., references [66–69]. According to Hüttel et al. [69], positive impact may be expected as a result of an increased availability of funds, which stimulate investments and input use. This results in the modernisation of productive capacity and acquisition of new production equipment enabling technological development [43], and thus helps enterprises to use economies of scale [68]. Further, the decrease in risk aversion of entrepreneurs is another theoretical cause of increased productivity as a result of subsidy-induced investment [65]. Henessey [67] identifies this phenomenon as a wealth effect of subsidies, because, by influencing wealth, entrepreneurs change their attitudes towards risk and thus expand their business more. Naturally, the positive effect of support on productivity may be caused by an increase in productive investments that would not be realised without support [31,44]. The above-mentioned process relates historically to the theory of the resource-based view (RBV) of the firm, suggesting that more (public in this case) resources may boost a firm's competitiveness and production efficiency [24].

The negative impact of investment support on productivity is mainly attributed to allocation inefficiencies. Firms adjust their investment decisions according to available support schemes and realise relatively less productive investments [31,44,48,63]. As stated by Špička [40], the investment support may lead to a deadweight loss when subsidised firms undertake a similar investment as they would realise without the subsidy. Rizov et al. [44] also point out that subsidies can cause technical inefficiency, as supported firms slacken off their efforts and willingness to seek cost-improving methods. In addition, there is also evidence on the crowding-out effect of investment support, e.g., references [70,71], where subsidised firms use public support to substitute for private investments. Finally, subsidies may be gained by less-productive enterprises due to erroneous identification of recipients by the support provider [48].

Regarding the focus of empirical studies on the effects of public support on the productivity of subsidized firms, one group of them deals with the regional support schemes intended to assist firms located in backward regions (disparities), or economically deprived "assisted areas", e.g., references [33,35,48,72,73]. These regional schemes aim to boost firm-level growth and productivity because more efficient (i.e., more productive) enterprises will be able to contribute to sustainable growth of these lagging regions [20]. Nevertheless, the findings show that public assistance improves productivity only partially. Bergström [48] found out the increase in productivity (measured by TFP) of subsidised firms in the first year after the subsidy; however, in the longer run, the productivity was decreasing. Ambiguous results were found out also by Harris and Robinson [35], who found that the regional support improved TFP only in some regions and industries. In their later study, Harris and Robinson [72] again evidenced negative TFP development of supported firms; however, regarding labour productivity, the impacts were positive. There is a consensus of the authors [33,48,72] that the assistance maintains less productive firms on the market, which limits the potential for the productivity growth and impedes the process of "creative destruction" described by Schumpeter [74].

Most other studies address productivity growth in conjunction with regional policies geared mainly to the manufacturing industries, e.g., references [25,32,34,43,75]. Such policies are regarded as an instrument to boost private investments with the aim to enhance growth, productive efficiency and competitiveness. Overall, most of the studies evidenced that public support fails to increase firms' productivity, e.g., references [34,44,45,76,77], and positive effects on productivity have been identified rather exceptionally, e.g., references [40,43,78].

Particular attention is paid to the impact of the Common Agricultural Policy (CAP) subsidies on farm productivity, e.g., references [31,44,49,53,79]. Although great attention has been paid to transition from coupled to decoupled subsidies and their effects on farm production and income, e.g., references [80–82], less focus has been devoted to the productivity effects of investment support, e.g., references [31,49,53]. Findings regarding the relationship between CAP subsidies and productivity are also inconsistent and mostly negative [49,82,83]. In contrast, Nilsson [31] discovered a positive effect of investment support on the productivity of small firms in Sweden, just as Medonos et al. [52] and Ratinger et al. [53] confirmed significant benefits regarding farms' labour productivity improvements in the Czech Republic.

We provide a review of quantitative studies specifically targeted at productivity changes induced by the investment support policies and summarise the empirical findings in Table 1, which presents important characteristics of particular studies, i.e., authors, country of analysis and the support programme, period of study, sample, aided industry, productivity variable, empirical approach and findings. Before we comment on the previous findings, we would like to point out that the previously published studies often lack fundamental information, such as a number of supported and non-supported firms, sample description and summary statistics. Such issues complicate the comparability of findings over time. We believe that this should be improved in the forthcoming studies.

**Table 1.** Overview of empirical studies investigating the effects of subsidies on firm-level productivity (only effects on productivity variables are reported).

| Authors | Country of Analysis, Programme | Period, Sample | Aided Sectors | Productivity Variables | Empirical Approach | Findings |
|---------|-------------------------------|----------------|---------------|------------------------|--------------------|----------|
| Bergström [48] | Sweden, Regional policy intended to mitigate regional disparities | 1987–1993, 76 supported firms, 884 non-supported firms | Manufacturing industry | TFP growth | OLS regressions and correlations | In the first year after support, productivity of subsidised firms increased. However, in the long run, the more subsidies a firm has received, the more TFP has decreased. |
| Harris and Robinson [35] | United Kingdom, Regional Selective Assistance (RSA) and SMART/SPUR grants | 1986–1998, 13,294 supported firms, 32,282 non-supported firms | Manufacturing industry | TFP | GMM estimates | The support improved the productivity of assisted firms compared with average level, though only in some regions and industries. |
| Harris and Robinson [72] | United Kingdom, Regional Selective Assistance (RSA) | 1990–1998, 57,419 supported firms, 90,088 non-supported firms | Manufacturing industry | TFP | Decomposition of TFP. Calculations and comparisons at sectoral and regional levels. | Decomposition of TFP suggested that firms supported by the scheme substituted capital for labour. Assisted plants reported higher labour productivity growth. However, regarding TFP assisted firms experienced negative growth. |
| Harris and Trainor [43] | Northern Ireland, Selective Financial Assistance (SFA) | 1983–1998, 436 supported firms/plants | Manufacturing industry | TFP | GMM estimates | The authors find a positive impact of SFA on the TFP of assisted firms. |
| Pellegrini and Centra [76] | Italy, Regional Policy, Law 488/92 | 1995–2001, 665 supported firms, 1493 non-supported firms | Manufacturing industry | Labour productivity | PSM + DID | The authors find that labour productivity in subsidised firms grew slower compared to non-supported firms. |
| Girma et al. [75] | The Republic of Ireland, Government grants | 1992–1998, 1087 supported firms/plants | Manufacturing industry | TFP | GMM estimates | The authors find a positive impact of the intervention on the TFP of assisted firms/plants. However, if the companies' debt ratio is more than 100, then it might lead to negative effects. |
| Bernini and Pellegrini [34] | Italy, Regional policy, Law 488/1992 | 1996–2004, 468 supported firms, 728 non-supported firms | Manufacturing industry | TFP | PSM + DID | The authors find a negative impact of Law 488 on the TFP of supported firms. |
| Mary [49] | France, Common Agricultural Policy (CAP) subsidies, Pillars 1 and 2 | 1996–2003, 1529 supported firms/farms | Agriculture (crop farms) | TFP | GMM estimates | The author finds no significant effect of investment subsidies on the farm's productivity. |

| Authors | Country of Analysis, Programme | Period, Sample | Aided Sectors | Productivity Variables | Empirical Approach | Findings |
|---|---|---|---|---|---|---|
| Rizov et al. [44] | EU-15 countries, various subsidies allocated through the Common Agricultural Policy (CAP) | 1990–2008, 488,275 observations across EU-15 countries, number of farms/firms is not documented | Agriculture (commercial farms) | TFP | Cross-country correlations before and after decoupling reform | The authors find a negative relationship between subsidies and the level of productivity, before the decoupling reform. Nevertheless, after the decoupling reform in 2003, the correlations for some countries changed to positive. |
| Moffat [33] | United Kingdom, (Scotland only), Regional Selective Assistance (RSA) | 1984–2005 number of plants/firms is not documented. There are 5010 treated and 45,455 non-treated observations | Manufacturing industry | TFP | PSM + GMM | The author estimated effects for firms with respect to high, medium-high, medium-low and low tech sectors. Results show a negative impact of RSA on the TFP of firms operating in medium-low and low tech sectors. |
| Criscuolo et al. [73] | United Kingdom, Regional Selective Assistance (RSA) | 1997–2004, 21,404 supported firms | Manufacturing industry | TFP | OLS and IV regressions | The authors fail to find any significant impacts of RSA on the firm's TFP. |
| Bernini et al. [32] | Italy (for regions South and Centre-North only), Regional policy, Law 488/92 | 1996–2007 641 supported firms, 1233 non-supported firms (in total) | Manufacturing industry | TFP | RDD | The authors find a negative effect on TFP growth in the short term. However, they find long-term positive effects after 3–4 years. |
| Busom et al. [78] | Colombia, Not specific (any) public support for innovation projects | Service sector: 2010–2011; 2373 firms (supported 95 firms); Manufacturing sector: 2009–2010; 905 firms (supported 72 firms) | Manufacturing and service industries | Labour productivity | 2SLS and Quantile regression estimates | The authors find positive effects of policies supporting innovations on firm's labour productivity. |
| Howell [45] | China, Industrial policy | 2001–2007, 6828 supported firms, 50,797 non-supported firms (in 2001), 16,495 supported firms, 96,171 non-supported firms (in 2007) | Manufacturing industry | TFP | PSM + RE | The author estimated effects for firms with respect to high, medium-high, medium-low and low tech sectors. The authors find a negative impact of subsidies on TFP in all sectors. |
| Nilsson [31] | Sweden, Rural Development Programme (RDP) | 2007–2013, 4601 supported firms, 27,899 non-supported firms | Agriculture | Labour productivity, TFP | CEM + DID, FE | The author finds a positive effect of RDP on firm-level productivity (both labour and TFP), but only for small firms. |

Notes: TFP—Total factor productivity, OLS—Ordinary least squares, GMM—Generalised method of moments, PSM + DID—Propensity score matching with difference in differences approach, PSM + GMM—Estimation of GMM based on matched sample, IV—Instrumental variables approach, RDD—Regression discontinuity design, 2SLS—Two-staged least squares, PSM + RE—Random effects regressions on matched sample, CEM + DID—Coarsened exact matching with difference in differences approach, FE—Fixed effects regressions.

Regarding the obtained empirical evidence, the findings are still ambiguous. Several studies report the positive impact of investment support on labour productivity [31,78] or TFP [43]; however, most of them found negative or no productivity effects, both on labour productivity [76] and TFP [33,34,45,49]. There are even studies reporting contradictory results in terms of labour productivity and TFP [72], and studies confirming only partial positive impact of investment support on productivity, e.g., in terms of time [48], due to the age of enterprises [75], or in terms of various regions and industries [35].

Given the inconclusive empirical evidence on the relationship between investment subsidies and productivity of supported enterprises (and concerns on comparability of the previously published studies), this research question remains open and topical. Therefore, this study on the productivity effects of EU public support responds to the need to conduct rigorous impact evaluations of support policies to ensure "smart, sustainable and inclusive growth", as targeted by the European Commission [20]. The following section describes our empirical approach and collected firm-level dataset.

## 3. Empirical Approach and Data

### 3.1. Empirical Approach and Tested Hypothesis

The objective of this study is to assess the effects of subsidies on the productivity of firms in the Czech food industry. Building on the previously introduced concepts of subsidy-induced investments (wealth effects of subsidies), economies of scale and resource-based view (RBV) of the firm, we assume the following hypothesis to be tested:

**Tested Hypothesis:** The firms supported by public grants from the OPEI reported after the end of intervention higher values of productivity indicators, compared to non-supported firms and compared to the period before the public intervention.

The most straightforward methodological approach towards the evaluation of public policies and intervention is when the aid is distributed towards recipients randomly. This means that there is no selection process. Effects of such interventions are assessed by a simple comparison of key indicators before and after the end of the intervention. Unfortunately, most of the public grants are not allocated randomly, and there exists some form of selection process (which is also a case of the OPEI). This means that some factors might influence the likelihood of obtaining a grant, and thus the simple comparison of indicators cannot be used, because it would be biased [84,85].

In the absence of random distribution of grants (policy aid), evaluation scholars, economists and empirical researchers from the field most often use the impact assessment instrumental variables (IV) approach, regression discontinuity design (RDD) and the propensity score matching (PSM) approach. Each of these methods has its pros and cons. For example, IV's approach requires a good set of instruments, RDD examines effects only on part of the programme's participants (close to cutting edge), and PSM's power stands on the availability of the observable characteristics of the participants and non-participants [84,85].

Therefore, no methodological approach is perfect, and we always need to bear in mind the limitations of each approach. Given the fact that we lack good instruments for the IV's approach (e.g., applications scores), and we would like to assess the programme as a whole (not just firms above and below the threshold), we follow the PSM approach, which was for this purpose also recommended by the European Commission's representatives [86]. In line with the previously published studies, e.g., references [24,30,31,39,87], we combine the PSM approach with a difference in differences approach (DID), which takes into account changes in outcomes compared to the period before the intervention. The intervention took place during 2007–2013 and, thus, we consider as pre-treatment period the years 2005–2006 and post-intervention period the years 2014–2015 (two-year periods). Unfortunately, newer data are still not available for the analysis, and thus this two-year period allows us to identify rather short-term effects of the intervention.

In the following section, we introduce the collected sample and all firm-level variables. Then we empirically estimate the third variable measuring firm-level productivity—TFP. The fourth section is dedicated to the implementation of the PSM approach, and it begins with the estimation of the logistic regression, which is needed for the calculation of the likelihood of being supported by the OPEI. The obtained score (probability of participation) is used to match both groups of companies based on the observable characteristics, using the three different matching techniques. Then we calculate the average difference between both groups of companies and as a difference between the periods before and after the end of the intervention (so-called average treatment effect on the treated—ATET) [84,85].

### 3.2. Sample and Variables

We base our analysis on the Czech food processing industry. According to the Ministry of Agriculture of the Czech Republic [88,89], there were on average 7542 companies operating in the industry during 2007–2013. The OPEI aimed to mainly support the competitiveness of SME companies. Only firms aiming to obtain support for projects realised in the capital of Prague were excluded from application process due to programme rules of the cohesion policy (for more details, see Dvouletý and Blažková [39]).

We have used a database of the CzechInvest [90] agency to identify recipients of subsidies within the industry (based on the CZ-NACE codes), and we found 203 firms that received public support within OPEI. Most of them (95.7%) were SMEs. The rest of the companies operating in the industry can, therefore, be considered as a control group. We have used business registration numbers to collect firm-level data from the database MagnusWeb [91], which includes balance sheets and profit-and-loss statements of the Czech enterprises. We have exploited data for the Czech food processing companies based on the CZ-NACE codes. Particular food processing sectors can be seen in Table 2. Because there were no supported companies from CZ-NACE 102, i.e., processing and preserving of fish and fish products, this sector was excluded from the counterfactual impact analysis. We have also tried to collect additional data from the websites of the Ministry of Justice of the Czech Republic [92] to minimise the missing values. Unfortunately, there are a lot of missing data, so the final dataset consists of 1381 firms in total—157 (ranging from 138 to 157 depending on the outcome variable; see Table 5) of them belong to the Treated group, i.e., subsidised firms identified using the database of CzechInvest [90], and the remaining 1224 (ranging from 799 to 1224 depending on the outcome variable; see Table 5) firms are considered as the control group. As there were 203 firms participating in the OPEI support within the Czech food processing industry [90], the data sample accounts for 77.3% of the programme participants within the sector. Table 2 shows the dataset distribution and structure concerning the sectors. We also define variables involved in the analysis in Table 3, and Table 5 provides summary statistics for all variables showing pre-intervention and post-intervention values of selected statistical indicators (mean, SD, min, max, number of observations).

Regarding the purpose of our study, i.e., effects of the investment support on the productivity of subsidised firms, we employed three productivity indicators as the outcome variables—Production Efficiency, Labour Productivity and TFP. Although the first two indicators can be easily calculated on the basis of data from the financial statements of firms, there are many parametric and semi-parametric techniques used for the estimation of TFP. Some of them rely on the calculation of indices, others on the regression estimates, e.g., references [93–96]. A very practical review of existing methods and approaches was elaborated by Van Beveren [93], who does not find one perfect estimator and proposes an implementation of multiple techniques and their empirical comparison concerning the particular sample.

Therefore, we have decided to estimate TFP by two techniques—by simple ordinary least squares (OLS) regression and by two-way generalised method of moments (GMM) regression. For the firm-level variables and estimation procedures, we follow Van Beveren [93]. We use total sales (*Total Sales*) as a proxy for output, and, as for the inputs, we use consumption of materials and services (*Material Consumption*), and personnel costs (*Personnel Costs*) together with the capital inputs (*Tangible Fixed Assets*). In line with

Van Beveren [93] (p. 100), we transform all variables into a form of natural logarithms, and we estimate the Cobb–Douglas production function represented by the following Equation (1), where *i* refers to firm, *t* to time, and *ε* to the time and producer specific deviation from the mean consisting of observable ($v_{it}$) and unobservable components ($u^q_{it}$)):

$$Total\ Sales_{it} = \beta_0 + \beta_1 Tangible\ Fixed\ Assets_{it} + \beta_2 Material\ Consumption_{it} + \beta_3 Material\ Consumption_{it} + \varepsilon_{it} \quad (1)$$

Then we calculate firm-level TFP ($\omega_{it}$) from the following Equation (2)—for details, please see Van Beveren [93] (p. 100):

$$\hat{\omega}_{it} = \hat{v}_{it} + \hat{\beta}_0 = Total\ Sales_{it} - \hat{\beta_1}Tangible\ Fixed\ Assets_{it} - \hat{\beta_2}Material\ Consumption_{it}$$
$$- \hat{\beta_3}Material\ Consumption_{it} \quad (2)$$

Following the approach described above, we estimate the above-presented equations with the traditional OLS regression and with the two-way dynamic GMM regression. The estimation results are presented in Table 4.

**Table 2.** Sample structure with respect to sector.

| NACE Code | Treated (N) | Freq. (%) | Control (N) | Freq. (%) | Total (N) | Freq. (%) |
|---|---|---|---|---|---|---|
| CZ-NACE 101 (Production, processing, preserving of meat) | 1 | 0.64 | 221 | 18.06 | 222 | 16.08 |
| CZ-NACE 102 (Processing and preserving of fish and fish products) | 0 | 0.00 | 11 | 0.90 | 11 | 0.80 |
| CZ-NACE 103 (Processing and preserving of fruit and vegetables) | 2 | 1.27 | 37 | 3.02 | 39 | 2.82 |
| CZ-NACE 104 (Manufacture of vegetable and animal oils and fats) | 4 | 2.55 | 5 | 0.41 | 9 | 0.65 |
| CZ-NACE 105 (Manufacture of dairy products) | 4 | 2.55 | 48 | 3.92 | 52 | 3.77 |
| CZ-NACE 106 (Manufacture of grain mill and starch products) | 10 | 6.37 | 55 | 4.49 | 65 | 4.71 |
| CZ-NACE 107 (Manufacture of bakery and farinaceous products) | 47 | 29.94 | 286 | 23.37 | 333 | 24.11 |
| CZ-NACE 108 (Manufacture of other food products) | 45 | 28.66 | 239 | 19.53 | 284 | 20.56 |
| CZ-NACE 109 (Manufacture of prepared animal feeds) | 7 | 4.46 | 98 | 8.01 | 105 | 7.60 |
| CZ-NACE 110 (Manufacture of beverages) | 37 | 23.57 | 224 | 18.30 | 261 | 18.90 |
| Total | 157 | 100.00 | 1224 | 100.00 | 1381 | 100.00 |

Note: N—no. of observations. Source: own elaboration.

**Table 3.** List of variables.

| Variable | Definition |
|---|---|
| **Treatment variable** | |
| Treated | Variable indicates whether the particular firm participated in the OPEI programme. |
| **Control variables** | |
| Year of Registration | Variable refers to the year when the company was officially established. |

**Table 3.** *Cont.*

| Variable | Definition |
|---|---|
| Legal Form | Variable divides firms into the four dummy categories according to their legal entity: freelancer/self-employed, company with limited liabilities, joint-stock company and other. |
| Company Size | Variable divides firms into the three dummy categories, according to the amount of employees reported: small (0–49 employees), medium (50–249 employees) and large (250 and more employees). |
| Region | Variable divides firms into the 14 NUTS III dummy categories according to the Czech region, where they operate (control group), esp. where they realised the support project (treated group). |
| Sector | Variable divides firms into the 10 NACE dummy categories according to their business activity. |
| Profit/Loss | Variable is calculated as an average pre-intervention (2005–2007) profit/loss. |
| Total Assets | Variable represents an average pre-intervention (2005–2007) value of firm assets. |
| Trade Margin | Variable is calculated as an average pre-intervention (2005–2007) difference between sales of goods and costs of goods sold. |
| Personnel Costs | Variable represents an average pre-intervention (2005–2007) personnel costs of a firm. |
| Debt Ratio | Variable is calculated as an average percentage share of liabilities of the firm and its assets during the years 2005–2007. |
| **Outcome variables** | |
| Production Efficiency | Variable is calculated as the ratio of sales and production consumption of the firm. Production–consumption involves all variable costs related to the production of goods and services, such as material and energy costs, except for labour costs. |
| Labour Productivity | Variable is calculated as the ratio of value added of the firm and its labour cost. |
| TFP | Variable is estimated by two techniques—by simple OLS regression and two-way GMM regression—with the use of Cobb–Douglas production function (see Equation (1)) and calculated from Equation (2) based on Van Beveren [93]. |

Note: The outcome variables are calculated as average values in two analysed periods, i.e., before intervention (during the years of 2005–2006) and after intervention (2014–2015). Source: own elaboration.

**Table 4.** Estimation of TFP with OLS and two-Way GMM (based on firm-level panel data from years 2005–2015).

| Estimation Technique | (1) Robust SE OLS | (2) Two-Way GMM |
|---|---|---|
| **Independent/Dependent Variables** | **Log(Sales)** | **Log(Sales)** |
| Log(Tangible Fixed Assets) | 0.00851 (0.00552) | 0.0185+ (0.0104) |
| Log(Personnel Costs) | 0.290 *** (0.0101) | 0.253 *** (0.0178) |
| Log(Material Consumption) | 0.696 *** (0.00910) | 0.726 *** (0.0165) |
| Constant | 1.022 *** (0.0357) | 0.894 *** (0.0714) |
| Observations | 11,430 | 11,430 |
| Wald chi2(3) | 35172.12 | 23804.01 |
| Prob > chi2 | 0.00 | 0.00 |

Note: Standard errors in parentheses: $+$ $p < 0.10$, * $p < 0.05$, ** $p < 0.01$, *** $p < 0.001$. Details for two-Way GMM Estimation: number of groups: 1828, lags of instruments: up to two-year lag, number of instruments: 121, Sargan test's *p*-value: 0.00. Source: own elaboration.

Finally, we apply the formula from Equation (2) to obtain firm-level values of TFP. The estimated values are summarised together with the remaining variables in Table 5. Both estimations of TFP

are tightly positively correlated (correlation coefficient = 0.9976) as expected and needed [93]. Thus, we use both variables as outcome variables for our impact evaluation presented in the next section.

**Table 5.** Average outcomes before and after the programme before the application of the matching procedures.

| | **Before the Programme (2005–2006)** | | | | | | | |
| --- | --- | --- | --- | --- | --- | --- | --- | --- |
| **Variable** | **Production Efficiency** | | **Labour Productivity** | | **TFP** (OLS) | | **TFP** (GMM) | |
| Group | Control | Treated | Control | Treated | Control | Treated | Control | Treated |
| Mean | 3.25 | 2.27 | 1.82 | 1.68 | 1.03 | 1.05 | 0.96 | 0.98 |
| SD | 20.33 | 2.26 | 6.41 | 1.39 | 0.50 | 0.45 | 0.50 | 0.46 |
| Min | −3.77 | 0.63 | −29.67 | −8.95 | −1.98 | 0.35 | −1.99 | 0.23 |
| Max | 555.21 | 16.33 | 134.65 | 9.29 | 5.16 | 3.13 | 5.21 | 3.05 |
| N | 920 | 145 | 854 | 140 | 799 | 138 | 799 | 138 |
| | **After the Programme (2014–2015)** | | | | | | | |
| **Variable** | **Production Efficiency** | | **Labour Productivity** | | **TFP** (OLS) | | **TFP** (GMM) | |
| Group | Control | Treated | Control | Treated | Control | Treated | Control | Treated |
| Mean | 3.42 | 2.30 | 1.73 | 1.81 | 1.03 | 1.05 | 0.98 | 0.98 |
| SD | 16.80 | 2.26 | 5.71 | 0.96 | 0.59 | 0.42 | 0.59 | 0.42 |
| Min | 0.00 | 0.79 | −32.5 | −0.43 | −2.91 | −0.08 | −2.94 | −0.17 |
| Max | 540.26 | 20.15 | 128.4 | 7.05 | 5.42 | 3.13 | 5.47 | 3.05 |
| N | 1224 | 157 | 1129 | 156 | 1033 | 156 | 1033 | 156 |

Note: N—no. of observations. Source: own calculations.

## 4. Results of the Counterfactual Impact Evaluation

We base the microeconomic assessment of the productivity effects of public subsidies on the PSM approach combined with a DID approach. We began the empirical analysis by the inspection of the outlier observations that might bias the results [97]. To detect outliers, we used the programme "bacon" implemented in STATA 14 software (for details see reference [98]) with parameters of financial variables (Total Assets, Personnel Costs, Total Sales and Profit/Loss). The programme detected 19 outlier observations among the non-supported firms that have been removed from the initial sample. We have also removed from the initial sample 11 control companies from the CZ-NACE 102 sector, because there were no treated companies (see Table 2).

As the next step, we estimate the logistic regression needed for the calculation of the propensity score, which would allow us to match both groups of enterprises (supported, i.e., Treated; and non-supported, i.e., Control) based on the collected observable characteristics [85]. After that, we apply the three different matching techniques (kernel, radius with calliper 0.01 and nearest neighbour with one nearest neighbour). Then we run the matching diagnostics to make sure that both groups of firms are not statistically different from each other, when it comes to the observable characteristics, e.g., references [99–101]. Finally, we estimate the ATET, calculated as a difference between periods after the end of the programme (2014–2015) and before the programme had started (2005–2006).

### 4.1. Estimation of the Propensity Score

We estimate the odds of participation in the programme (propensity score) with the help of logit model (dependent variable: Treated = 1). It was very important to select the right independent firm-level characteristics (covariates) for the model to make sure that we accounted for the heterogeneity across both groups of firms to fulfil so-called conditional independence assumption (CIA) or unconfoundedness, e.g., references [102,103]. The selection of the variables therefore significantly affects the power of the PSM approach. Khandker et al. [85] suggest including variables having an impact on the success of the programme's application and other variables highlighted by previous research and theory. Therefore, our logistic regression covers the standard firm-level characteristics and determinants of profitability [22,23,31,104–106] that were available in our data. We control for Year of Registration, Legal Form, Company Size, Region, Sector and pre-intervention financial indicators

(Profit/Loss, Total Assets, Trade Margin, Personnel Costs and Debt Ratio). The results of the logistic regression that were estimated with the robust standard errors are presented in Table 6. We keep in our estimates both significant and non-significant independent variables (covariates) to obtain the most precise estimate of the propensity score that will be used for matching subsequently. The most statistically significant covariates in the logistic regression were found to be the Sector and Debt Ratio of the firms.

**Table 6.** Robust logistic regression applied for the calculation of the propensity score.

| Independent Variables/Dependent Variable | TREATED = 1 |
|---|---|
| Year of Registration | 0.00418 (0.0261) |
| Self-employed/Freelancer | . . |
| Limited Liabilities Company | 0.189 (1.018) |
| Joint Stock Company | 0.246 (0.893) |
| Other | . . |
| Size Micro (0–10 Employees) | . . |
| Size Small (10–49 Employees) | 0.419 (0.996) |
| Size Medium (50–249 Employees) | 1.444 (0.901) |
| Size Large (250+ Employees) | . . |
| Region Prague | . . |
| Region South Moravia | −0.359 (0.794) |
| Region South Bohemia | −0.301 (0.986) |
| Region Karlovy Vary | 1.453 (1.073) |
| Region Vysocina | −0.0320 (0.774) |
| Region Hradec Kralove | −0.337 (0.874) |
| Region Liberec | 0.367 (1.411) |
| Region Moravia-Silesia | 0.699 (0.810) |
| Region Olomouc | 0.500 (0.824) |
| Region Pardubice | 0.776 (0.955) |
| Region Pilsen | 0.0984 (0.972) |
| Region Central Bohemia | 0.212 (0.759) |
| Region Zlin | 0.300 (0.816) |
| Region Usti nad Labem | . . |
| Production, processing, preserving of meat (CZ-NACE 101) | . . |
| Processing and preserving of fish and fish products (CZ-NACE 102) | . . |
| Processing and preserving of fruit and vegetables (CZ-NACE 103) | −2.480 *** (0.594) |
| Manufacture of vegetable and animal oils and fats (CZ-NACE 104) | 1.580 * (0.719) |

**Table 6.** *Cont.*

| Independent Variables/Dependent Variable | TREATED = 1 |
|---|---|
| Manufacture of dairy products (CZ-NACE 105) | −2.716 *** <br> (0.631) |
| Manufacture of grain mill and starch products (CZ-NACE 106) | -0.639 <br> (0.737) |
| Manufacture of bakery and farinaceous products (CZ-NACE 107) | −0.799+ <br> (0.412) |
| Manufacture of other food products (CZ-NACE 108) | 0.259 <br> (0.324) |
| Manufacture of prepared animal feeds (CZ-NACE 109) | −1.284 * <br> (0.600) |
| Manufacture of beverages (CZ-NACE 110) | . <br> . |
| Profit/Loss (2005–2006) | 0.00000496 <br> (0.00000935) |
| Total Assets (2005–2006) | −0.00000223 <br> (0.00000176) |
| Trade Margin (2005–2006) | 0.0000124 <br> (0.0000138) |
| Personnel Costs (2005–2006) | 0.0000173 <br> (0.0000121) |
| Debt Ratio (2005–2006) | −0.00572+ <br> (0.00343) |
| Constant | −10.13 <br> (51.83) |
| Observations | 530 |
| Wald chi2(38) | 511.28 |
| Prob > chi2 | 0.00 |
| Pseudo R2 | 0.177 |
| AIC | 526.8 |
| BIC | 655.0 |

Note: AIC = Akaike Information Criterion, BIC = Bayesian information criterion estimates are based on pre-intervention firm-level characteristics. Standard Errors are in parentheses, statistical significance is reported as follows: + $p < 0.10$, * $p < 0.05$, ** $p < 0.01$, *** $p < 0.001$, omitted refers to a reference category or to a category with no observations. Source: own calculations.

## 4.2. Application of the Matching Techniques and Matching Quality Diagnostics

We do not rely only on one matching approach, and thus we match Treated and Control firms based on three different matching algorithms/techniques. We implement the current version (October 2018) of Leuven and Sianesi's PSMATCH2 commands. We use radius matching (with a calliper of 0.01) that allows determining the distance in propensity scores between both groups [107], nearest neighbour matching (with one nearest neighbour) that matches groups based on the closest propensity score [100], and nonparametric kernel matching technique which should be more consistent, when it is difficult to find close matches [85].

Then we assess the quality of the matching based on the statistical testing of the differences between both groups, distribution of the propensity score, mean and median bias before the implementation of the PSM and after it. We show the distributions of the standardised percentage bias and propensity scores, for both groups before and after the application of the matching procedures in appendices (Appendices A–C). Table 7 shows results of the statistical testing of the differences before and after the application of the matching techniques (regarding mean and median bias, Pseudo $R^2$ and LR chi$^2$). We may say that there are no statistically significant differences between both groups based on the observable characteristics, and the matching procedures have significantly contributed to the reduction of bias between both groups. Therefore, we believe that our selection of firm-level variables fulfils the CIA assumption. Thus, we might proceed towards the estimation of the ATET.

**Table 7.** Matching quality diagnostics.

| Matching Technique | Sample | Ps R$^2$ | LR chi$^2$ | p > chi$^2$ | Mean Bias | Median Bias |
|---|---|---|---|---|---|---|
| Kernel | Unmatched | 0.175 | 84.70 | 0.00 | 14.9 | 13.4 |
| Kernel | Matched | 0.048 | 13.88 | 1.00 | 6.5 | 6.1 |
| Radius with Caliper (0.01) | Unmatched | 0.175 | 84.70 | 0.00 | 14.9 | 13.4 |
| Radius with Caliper (0.01) | Matched | 0.048 | 13.88 | 1.00 | 6.5 | 6.1 |
| Nearest Neighbour (1) | Unmatched | 0.175 | 84.70 | 0.00 | 14.9 | 13.4 |
| Nearest Neighbour (1) | Matched | 0.049 | 13.31 | 1.00 | 7.2 | 3.8 |

Source: own calculations.

### 4.3. Estimated Average Treatment Effect on the Treated (ATET)

The final effect of the intervention on the firm productivity (ATET) is estimated as a difference between both groups (Treated and Control) and periods after the end of the programme (2014–2015) and before the programme had started (2005–2006). We have replicated the final estimates (reported in Table 8) by a hundred times and we have also used the common support option to obtain more robust results [108,109].

**Table 8.** Estimated Average Treatment Effect on the Treated (ATET) as a DID ((average outcomes 2014; 2015) – (average outcomes 2005; 2006)).

| Outcome Variable | Matching Technique | ATET | Std. Error | P > abs. Z | N |
|---|---|---|---|---|---|
| Production Efficiency | Kernel | 0.157 | 0.421 | 0.71 | 424 |
| Production Efficiency | Radius with Caliper (0.01) | −0.082 | 0.296 | 0.78 | 424 |
| Production Efficiency | Nearest Neighbour (1) | −0.492 | 0.365 | 0.18 | 424 |
| Labour Productivity | Kernel | 0.184 *** | 0.051 | 0.00 | 406 |
| Labour Productivity | Radius with Caliper (0.01) | 0.148 | 0.098 | 0.13 | 406 |
| Labour Productivity | Nearest Neighbour (1) | 0.315 * | 0.131 | 0.02 | 406 |
| TFP (OLS) | Kernel | −0.023 *** | 0.006 | 0.00 | 387 |
| TFP (OLS) | Radius with Caliper (0.01) | −0.036 * | 0.016 | 0.02 | 387 |
| TFP (OLS) | Nearest Neighbour (1) | −0.002 | 0.078 | 0.98 | 387 |
| TFP (GMM) | Kernel | −0.029 * | 0.014 | 0.03 | 387 |
| TFP (GMM) | Radius with Caliper (0.01) | −0.043 * | 0.018 | 0.02 | 387 |
| TFP (GMM) | Nearest Neighbour (1) | −0.007 | 0.049 | 0.89 | 387 |

Note: Statistical significance: * $p < 0.05$, ** $p < 0.01$, *** $p < 0.001$. Bootstrapped standard errors with 100 replications were used for all estimates together with common support option. Source: Own calculations.

We begin with the interpretation of the results for the variable measuring the overall production efficiency (Production Efficiency). Unfortunately, we do not find any statistically significant results for this variable, and we also do not see a consensus among the different matching techniques. Radius and nearest neighbour matching techniques indicate a negative impact of the programme, contrary to kernel matching, which indicates a positive coefficient. When it comes to labour productivity (Labour Productivity), all three matching techniques indicate a positive impact of the intervention. Nevertheless, only two of the three techniques found the effect to be statistically significant, namely, kernel and nearest neighbour. However, we believe that it is enough to provide a consensus across the matching techniques, especially in case the signs are not different from each other. Moreover, two of the three techniques show a statistically significant impact. We may see a similar picture when it comes to the TFP (estimated by both OLS and GMM), where all six coefficients indicate a negative effect of the intervention. However, only two estimated coefficients, for each TFP variable, out of three were found to be statistically significant.

To summarise the obtained findings, the matching estimates show that two years after the end of the intervention, the firms participating in the programme had lower values of TFP and higher values of labour productivity, compared to the non-supported companies. Nevertheless, we failed to prove any conclusive impact of the programme on the production efficiency. Thus, given our empirical findings, we cannot empirically support the tested hypothesis.

## 5. Discussion and Conclusions

Although policymakers generally expect entrepreneurship and SME policies to stimulate competitiveness and boost regional economic growth through investment subsidies, there is still an absence in the policy debate about the effectiveness of these public interventions paid from the taxpayer's money. Scholars point out [43,65,73] that investments in innovation activity, modernisation of productive capacity and acquisition of new production equipment enabling technological development may be justified only if they result in higher economic and technical efficiency (i.e., productivity). In fact, if an investment aid does not lead to productivity gains, one can regard such a support policy as failing in the long term.

However, to make adequate conclusions about the effectiveness of the public support, it is necessary to evaluate the specific support programmes implemented in different regions, considering the local conditions. Such evaluations might serve as an important feedback for the future policy adjustments [23,110,111]. We have reviewed the recently published quantitative studies analysing the productivity effects of public interventions, and we conclude that the findings are still ambiguous. Moreover, we would like to point out that the previously published studies often lack fundamental information, such as a number of supported and non-supported firms, sample description and summary statistics. Such issues complicate the comparability of the research over time. We believe that it is very important to conduct rigorous and transparent empirical evaluations to educate the research community, evaluators and policymakers. In addition, inconsistent empirical results can also arise due to the limited data on the support process and insufficient availability of comprehensive firm-level data for both subsidised firms and their comparable counterparts. Any efforts resulting in accessing data for conducting empirical evaluations need to be encouraged.

Therefore, our paper has empirically addressed this issue utilising the firm-level data on the Czech food processing firms to investigate the effectiveness of one specific public intervention. We conducted an empirical evaluation of the firms supported within the OPEI, a support scheme financed by the EU Cohesion Policy in 2007–2013. We focused solely on the productivity changes (by employing three key productivity indicators, i.e., production efficiency, labour productivity and TFP) induced by the subsidy, as the main goal of the OPEI was to promote competitiveness and growth [50], which cannot be achieved without productivity gains. From the methodological point of view, we have applied the counterfactual impact analysis (quantitative approach) to conduct a rigorous micro-econometric evaluation, which may give feedback for the policymakers when designing the interventions for the future programming period. Our analysis is focused on one sector only, and our dataset includes firms supported within the industry (157 firms, i.e., 77.3% of grant recipients) and a control sample of 1224 firms that have not been supported by the intervention.

The findings obtained from our study are twofold. First, the results show that the investment support positively affected labour productivity of subsidised firms, which signals positive effects of the programme, as in the previous studies by Blažková and Dvouletý [39], Špička [40] or Ratinger et al. [53]. This may be reflective of increased funds in aided firms [31,68], especially for small enterprises with lower capital, represented in the Czech food processing industry in large numbers [40,97]. Companies have thus acquired capital to buy new machines and equipment that have enabled more efficient production and led to increased productivity reflected in the labour productivity indicator. However, the TFP of these supported companies did not positively change during the period, and the counterfactual analysis confirmed a negative statistically significant effect of the subsidy on supported firms when compared with their non-supported counterparts.

In this respect, the findings show rather negative effects of the support programme on the productivity, as confirmed by most previous studies using TFP as a productivity variable when assessing the overall effects of the investment support, e.g., references [34,44,45,49]. Based on these results we assume that productivity of subsidised firms did not improve through an internal increase in efficiency. However, the positive impact on productivity was caused by the replacement of labour by other inputs rather than by a significant increase in the efficiency of the use of inputs, which indicates

no significant technological change. Our findings suggest that investments in modernisation and innovation in the Czech food processing industry were responsible for labour productivity growth rather than the overall firm-level efficiency improvements. Given the fact that TFP reflects the productivity changes that are not attributable to any production inputs but to the improvements in the combination of inputs used in production, e.g., technological development [31], investments induced by subsidy resulting in a negative impact on the firms' TFP mean that such investments do not spur technological development and thus economic growth. It prefigures our conclusion that, from the perspective of achieving the OPEI targets, the required effects of the support have not been fully achieved in the Czech food processing industry.

Several explanations for the productivity losses and consequent implications for refining of current support policy may be given. As productivity growth in a given sector can be achieved through investment or restructuring, non-supported firms are likely to have achieved productivity gains through restructuring, whereas supported firms have invested more and increased production [76]. At the same time, Bresnahan et al. [112] argue that the implementation of a labour-saving or cost-reducing technology does not always lead to productivity gains because of the lack of organisational structures or management methods aimed at facilitating the adoption of new technologies. Also, Edwards et al. [113] emphasise the need for skilled workers and managers capable of sustaining and leading change towards productivity gains. The results obtained from this study may also indicate that the support maintained less productive enterprises in the sector and thus limits the potential benefits in line with the process of "creative destruction" [74]. As suggested by some authors, e.g., those listed in references [31,48,82], the investment subsidy changes firm behaviour in the sense of lower motivation and effort, which results in a negative effect on firm productivity. Holmström [114] mentions a rent-seeking behaviour, i.e., firms may choose to reallocate productive resources to the process of seeking subsidies. To limit this effect, Terjesen et al. [111] and Audretsch and Link [115] suggest allocating resources to elite programmes with high-growth potential only or investing to boost entrepreneurial ecosystems, from which all entrepreneurs might benefit. If the support is not directed to the projects with growth potential, the main effect of such grant schemes will be that subsidised enterprises will be larger rather than more efficient [32]. Therefore, the sustainability and growth potential of the project proposals should be carefully assessed by the authorities responsible for granting the aid. Finally, given the fact that our results should be interpreted as revealing the short-term effects of subsidies on productivity, one may expect that some of the subsidized firms will go on in the long term to generate larger productivity gains leading to social-welfare benefits and positive market and technological spillovers, as supposed by Howell [45]. Therefore, this empirical evaluation should be repeated in the long term, and the results of short-term and long-term effects on productivity should be compared. We also need to acknowledge the missing firm-level data in the industry which reduced our sample size. Finally, there might also be other factors that might potentially influence the participation in the programme that we did not observe in our data, such as education and skills of the management of the companies.

The conducted analysis has shown how important it is to assess public interventions targeted at firm level, not only when it comes to the traditional measures of profitability, sales and growth, but also concerning the technological efficiency which may be measured with productivity indicators, such as TFP. Probably, this indicator is not often used in empirical evaluations so often, because it requires additional calculations and estimations. Thus, we would like to encourage future researchers and evaluators to include TFP in their analyses, and we recommend for this purpose a practical review written by Van Beveren [93].

We also believe that it would be very appropriate to extend the time period of the analysis as more data become available, and to investigate the effects of investment subsidies on firms' productivity in the long term (3–4 years after the end of programme), because benefits from such innovation-targeting schemes are likely to be expected over a longer time frame [35,116,117]. The empirical analysis could also include a broader set of firms across different industries and regions to contribute to the existing knowledge in a more general context. Another direction of research might be to decompose the change

in TFP into the efficiency change, economies of scale and technological progress as three possible kinds of productivity improvements [47], which would allow identifying in more detail particular channels through which investment subsidies affect firm productivity and thus long-term growth.

**Author Contributions:** Both authors have contributed equally to work reported.

**Funding:** This work was supported by the Internal Grant Agency of Faculty of Business Administration, University of Economics in Prague, under Grant No. IP300040 and by the Internal Grant Agency of Faculty of Regional Development and International Studies, Mendel University in Brno, under Grant No. 2019/005.

**Acknowledgments:** The authors thank all eight anonymous referees for their contributions to the development of this paper.

**Conflicts of Interest:** The authors declare no conflict of interest.

## Appendix A

Kernel Matching Technique Diagnostics Graphs (distributions of the standardised % bias on the top and propensity scores on the bottom, for both groups before and after the application of the matching procedures).

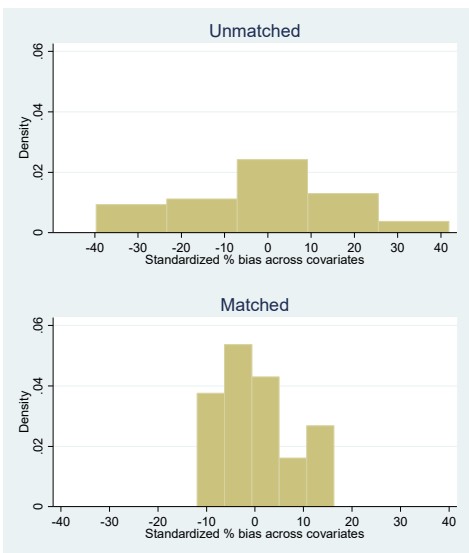

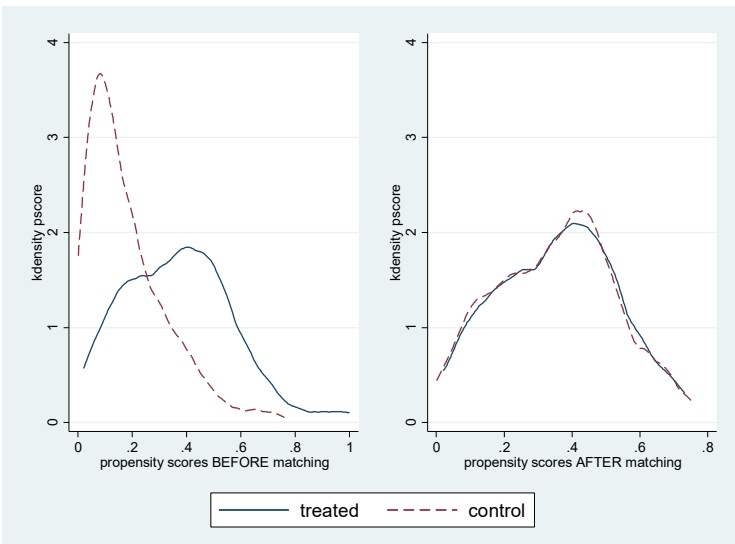

**Figure A1.** Source: own calculations.

## Appendix B

Radius with Caliper (0.01) Matching Technique Diagnostics Graphs (distributions of the standardised % bias on the top and propensity scores on the bottom, for both groups before and after the application of the matching procedures).

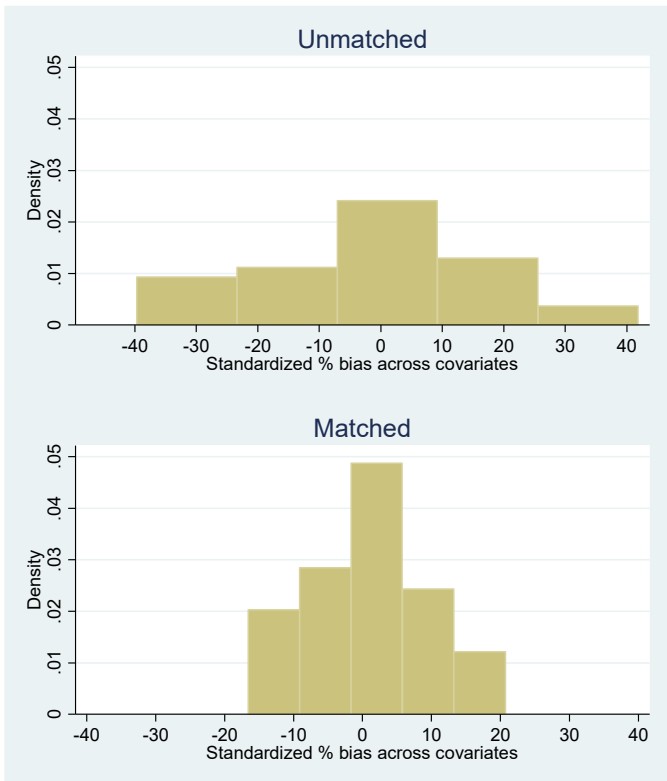

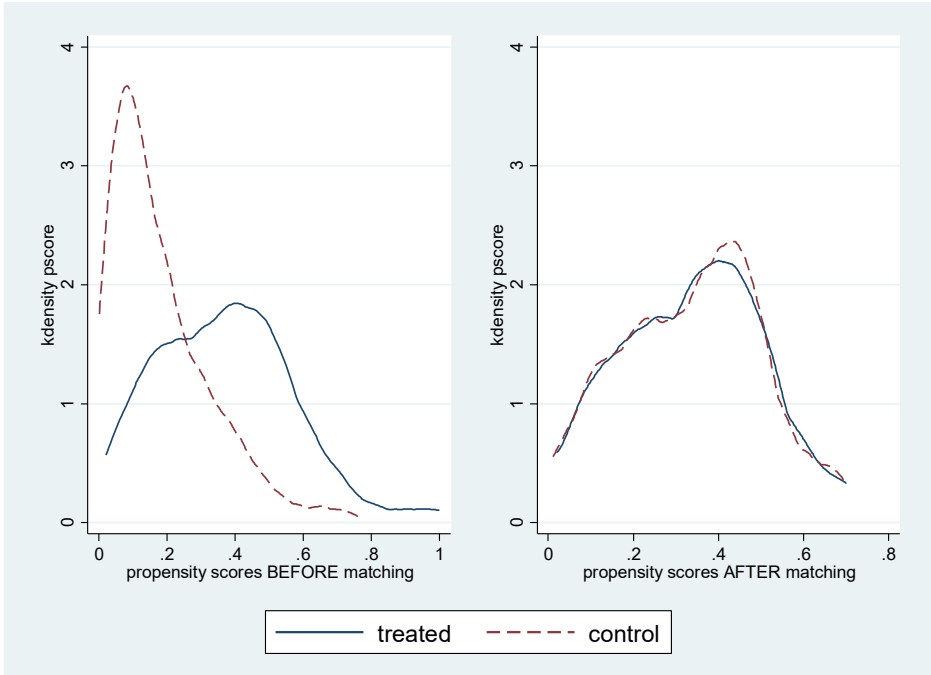

**Figure A2.** Source: own calculations.

## Appendix C

Nearest Neighbour (1) Matching Technique Diagnostics Graphs (distributions of the standardised % bias on the top and propensity scores on the bottom, for both groups before and after the application of the matching procedures).

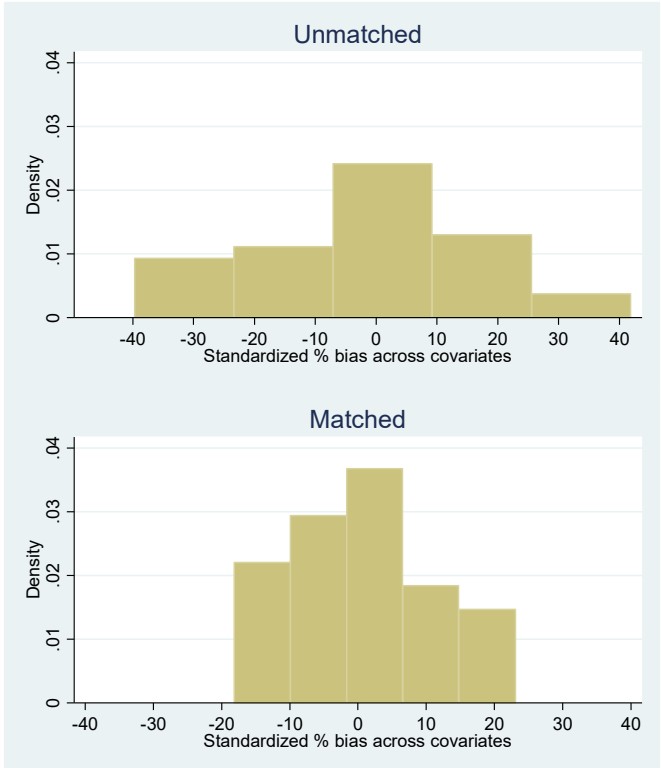

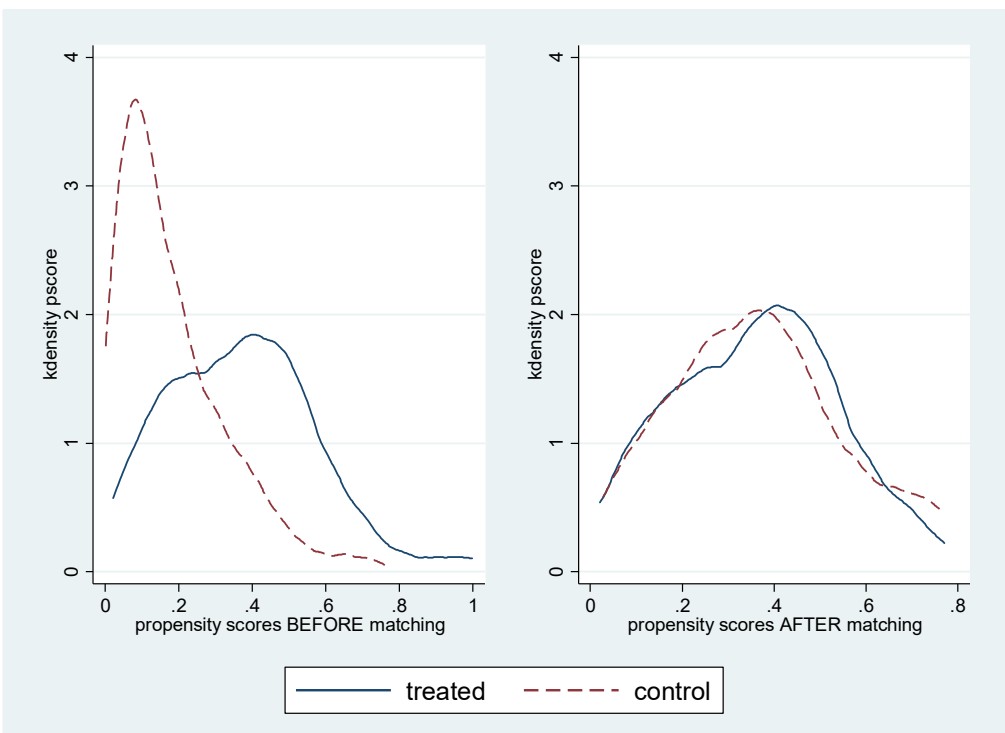

**Figure A3.** Source: own calculations.

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
