# Peer review of "The Impact of Public Grants on Firm-Level Productivity: Findings from the Czech Food Industry"

_sustainability, doi:10.3390/su11020552_

Reviewer 1 Report

Dear Authors,

Thank  you for the opportunity to review your article. I would like to  congratulate the authors for addressing such a relevant and difficult  issue regarding business subsidies. This is an important practical and  systemic issue in the design of modern capitalism and economic policy,  which deserves evidence-based attention and decision-making practice.

I  must say, the analysis performed looks incredibly complex, and it was  hard for me to fully comprehend the rationale or the various steps  taken, not being fully familiar with this part of the productivity  literature.

Let me try and confirm my understanding of the analytical procedure:

*  Firm level productivity metrics are estimated

* DID method is applied to test the difference in productivity growth between subsidized and non-subsidized firms

Departing  from this understanding, the article seeks an answer to the question,  whether subsidies have a positive productivity effect or not, in other  words the productivity growth among subsidized firms is greater than  those of non-subsidized firms, controlling for a range of control  customary variables.

The authors present this situation as that of  a quasi-experimental research design, where firms not receiving the  subsidies are considered a control group. This raises the question,  whether the selection criteria used to determine the eligibility for  subsidies has any effect on productivity growth at all. The authors use a  method that builds on observable characteristics of the firms, to  control for certain influences on the difference. However, in my view,  it requires a more explicit justification as to whether a randomized  control trial experimental setup is necessary at all for this kind of  analysis, and what the potential effect of this selection bias is on the  outcomes, or how these biases are mitigated during the analysis  performed.

In terms of the key issue examined in the article, the  discussion leads to question the real cause for successful programs. The  authors make the recommendation that careful selection of participation  in subsidy programmes can improve outcomes. This links back to the  question whether selection criteria – and the goodness of their  application – actually has more influence on the effectiveness of the  subsidies, than the actual impact that the subsidies had. This issue is  similar to that of the business excellence and quality award programmes.  Literature on those has been highly questioned because it was not clear  whether the outcomes were due to the businesses doing something better,  or because they engaged in competing for a programme and the systematic  procedure actually improved their performance.

So, what I recommend in the limited context of my understanding the method:

1.      Provide  a clear explanation to the analytical procedure and why the broad  steps/directions are taken to drive the results. The method is very  sophisticated and the methodologically less endowed reader may benefit  from the broad explanation why is what done, before indulging in the  detail.

2.      Ensure  to discuss the issue of experimental design and how the selection  process/criteria may influence the outcomes, and how these are being  controlled for. You state that ‘anyone’ was eligible for the programme  on page 7 / line 227. Was there any pattern recognisable that may have  had an influence? Was this controlled for?

3.      The  paper departs from a practical problem. This issue is around the  business subsidies. How do the results inform theory around this  problem? How can you best articulate the theoretical hook? I recommend  the authors to consider emphasising something along the lines of a  broader domain, such as ‘varieties of capitalism’, ‘evidence based  policy making’, ‘sustainability’ or ‘resource based view’ to  contextualise the study and integrate the conclusions into the a  systematic effort to progress theory. I would like to leave it up to the  authors to select a broader conceptual field, it does not have to be  what I noted before, but I believe the article would benefit from an  adjustment in this fashion.

Finally, I would like to draw the authors’ attention to the following typo: p. 9., live 321 – I believe it should be ‘ambiguous’

Thank you again for the paper, I certainly recommend publishing, with some minor adjustments as recommended above.

Authors' response:

Dear Reviewer,

Thank you very much for the time and efforts you invested in our manuscript, and we hope that we have been able to address all your comments. We have revised our manuscript according to your comments, and we explain the changes in light of comments below. All changes in the manuscript are tracked.

Reviewer´s Comments:

Dear Authors,

Thank you for the opportunity to review your article. I would like to congratulate the authors for addressing such a relevant and difficult issue regarding business subsidies. This is an important practical and systemic issue in the design of modern capitalism and economic policy, which deserves evidence-based attention and decision-making practice.

I must say, the analysis performed looks incredibly complex, and it was hard for me to fully comprehend the rationale or the various steps taken, not being fully familiar with this part of the productivity literature.

Let me try and confirm my understanding of the analytical procedure:

*  Firm level productivity metrics are estimated

* DID method is applied to test the difference in productivity growth between subsidized and non-subsidized firms

Departing from this understanding, the article seeks an answer to the question, whether subsidies have a positive productivity effect or not, in other words the productivity growth among subsidized firms is greater than those of non-subsidized firms, controlling for a range of control customary variables.

The authors present this situation as that of a quasi-experimental research design, where firms not receiving the subsidies are considered a control group. This raises the question, whether the selection criteria used to determine the eligibility for subsidies has any effect on productivity growth at all. The authors use a method that builds on observable characteristics of the firms, to control for certain influences on the difference. However, in my view, it requires a more explicit justification as to whether a randomized control trial experimental setup is necessary at all for this kind of analysis, and what the potential effect of this selection bias is on the outcomes, or how these biases are mitigated during the analysis performed.

In terms of the key issue examined in the article, the discussion leads to question the real cause for successful programs. The authors make the recommendation that careful selection of participation in subsidy programmes can improve outcomes. This links back to the question whether selection criteria – and the goodness of their application – actually has more influence on the effectiveness of the subsidies, than the actual impact that the subsidies had. This issue is similar to that of the business excellence and quality award programmes. Literature on those has been highly questioned because it was not clear whether the outcomes were due to the businesses doing something better, or because they engaged in competing for a programme and the systematic procedure actually improved their performance.

So, what I recommend in the limited context of my understanding the method:

1.      Provide a clear explanation to the analytical procedure and why the broad steps/directions are taken to drive the results. The method is very sophisticated and the methodologically less endowed reader may benefit from the broad explanation why is what done, before indulging in the detail.

Response:

We would like to thank the reviewer for a very nice summary of what we did and how and we confirm that his/her description is very much correct. We also understand his/her comments needing more guidance for readers are probably not fully familiar with this kind of research approach. We have substantially revised the section 3.1 Empirical Approach and Tested Hypothesis and we did our best to guide readers throughout the forthcoming sections of the paper that are focused on implementation of the counterfactual impact assessment.

Reviewer:

2.      Ensure to discuss the issue of experimental design and how the selection process/criteria may influence the outcomes, and how these are being controlled for. You state that ‘anyone’ was eligible for the programme on page 7 / line 227. Was there any pattern recognisable that may have had an influence? Was this controlled for?

Response:

We would like to thank the reviewer again for his/her clarification. By conducting logistic regression (Table 6), we calculate the propensity score (likelihood of being supported), which is regressed on all available firm-level characteristics we have (as very carefully described in section 4.1), and as we already written in the previous version of our paper

“We control for Year of Registration, Legal Form, Company Size, Region, Sector and pre-intervention financial indicators (Profit/Loss, Total Assets, Trade Margin, Personnel Costs and Debt Ratio).”

By estimating the probability of being supported we match (make twin firms), based on these characteristics mentioned above. The matching quality diagnostics reported in Table 7 show that there are no statistical differences between both groups when it comes to all above-mentioned characteristics. In other words, we managed to match for each of the supported companies a non-supported partner company with similar characteristics.  Moreover, in line with reviewer´s recommendation we have added a sentence stating that “The most significant covariates were found to be in the logistic regression the Sector and Debt Ratio of the firms. “

Khandker, S. R., Koolwal, G. B., Samad, H. A. Handbook on impact evaluation: quantitative methods and practices; World Bank Publications, 2010. DOI: https://doi.org/10.1596/978-0-8213-8028-4.

Reviewer:

3.      The paper departs from a practical problem. This issue is around the business subsidies. How do the results inform theory around this problem? How can you best articulate the theoretical hook? I recommend the authors to consider emphasising something along the lines of a broader domain, such as ‘varieties of capitalism’, ‘evidence based policy making’, ‘sustainability’ or ‘resource based view’ to contextualise the study and integrate the conclusions into the a systematic effort to progress theory. I would like to leave it up to the authors to select a broader conceptual field, it does not have to be what I noted before, but I believe the article would benefit from an adjustment in this fashion.

Response:

We would like to thank the reviewer for a constructive comment. We have tried to describe reasons for both, positive and negative impacts of public subsidies through the concepts of technological efficiency, economies of scale and wealth effects of subsidies when it comes to the positive impacts of subsidies and we have also explained potential reasons for the negative effects of public interventions in light of the allocation inefficiencies and crowding out effects of the subsidies.

Nevertheless, to be even more straightforward, we added to the theoretical section concept of resource-based view (RBV) of the firm, which assumes that more resources may boost firm´s competitiveness and production efficiency.

Reviewer:

Finally, I would like to draw the authors’ attention to the following typo: p. 9., live 321 – I believe it should be ‘ambiguous’

Response:

We would like to thank the reviewer for detecting the typo. We have corrected it.

Thank you again for the paper, I certainly recommend publishing, with some minor adjustments as recommended above 

Regards,

A reviewer

Reviewer 2 Report

Please refer to:

Bryła P., The impact of EU accession on the marketing strategies of Polish food companies, British Food Journal, 2012, Volume 114, Issue 8, pp. 1196-1209

Language issues:

line 181 - it should be "ambiguous"

190 - question

300 - which

464 - has shown how important it is

Authors' response:

We would like to thank the reviewer for detecting the typos and for suggesting a new reference. We cite the proposed paper, and we corrected the detected typos and language issues.

Reviewer 3 Report

Thanks for the opportunity of reading this interesting research.

I have several doubts about if this article fits with the scope of the journal, since no mention to sustainability or CSR is included in the manuscript since the main aim is to analyze if public grants affect firm´s productivity in the context of an specific industry in Czech Republic. It seems to be a manuscript of Public Administration more than Sustainability.

The contribution of the paper should be a little bit more clear, as well as, the number of references should be shorter. For example, there are lots of brackets with more than 3 references (one with more than 10). As a recommendation, the most significant references should be included and if all are relevant, their contribution should be pointed out specifically.

The theoretical framework should include the most significant theories that support the positive/negative effect of public grants on firm´s productivity, in addition to the discussion of the empirical results from previous research. No hypotheses are included.

Regarding to the sample, I would like to say that the results could be really bias due to the imbalance in the sample distribution. In order to overcome this point, several researchers create the control group with a similar number of companies with a similar profile (size, subsector, performance...) of the group with public support. More information about the companies included in the sample should be reported.

It is surprinsing that in Table 3 appears the possibility that some companies are large ones (+250 employees). Considering that all the theoretical discussion included in the paper is about SMEs, this fact should be considered.

Regarding to Logistic Regression, there is too many independent variables which means that there may be a collinearity problem. Authors should consider the convenience of summarizing the impact of at least region and subsector variable, that is, finding a point in the literature that help them to divide the sample into less subgroups in both variables. 

Authors' response:

Dear Reviewer,

Thank you very much for the time and efforts you invested in our manuscript, and we hope that we have been able to address all your comments. We have revised our manuscript according to your comments, and we explain the changes in light of comments below. All changes in the manuscript are tracked.

Reviewer´s Comments:

Thanks for the opportunity of reading this interesting research.

I have several doubts about if this article fits with the scope of the journal, since no mention to sustainability or CSR is included in the manuscript since the main aim is to analyze if public grants affect firm´s productivity in the context of an specific industry in Czech Republic. It seems to be a manuscript of Public Administration more than Sustainability.

Response:

Thank you very much for raising this issue. We would like to convince the reviewer that the paper has at least two fits concerning the aims and scope of the Sustainability journal. The first explanation is offered already in the “Theoretical Background” section. We explain that cohesion policies aim to

“aim to boost firm-level growth and productivity because more efficient (i. e. more productive) enterprises will be able to contribute to sustainable growth of these lagging regions [20].” 

and we also try to explain in a very careful way throughout the text that firm-level sustainability (and especially the long-term sustainability) is associated in the literature (and operationalised) regarding productivity indicators. We also explain that firm-level productivity is very tightly linked with concepts like efficiency, economies of scale and technological improvements.

Second, we would like to point out that our manuscript was submitted to the special issue “Researching Entrepreneurship at Different Levels: Micro, Meso and Macro” available at https://www.mdpi.com/journal/sustainability/special_issues/Meso_and_Macro, which encourages authors to contribute with the papers focused on “Entrepreneurship policies and their evaluation”.

Reviewer:

The contribution of the paper should be a little bit more clear, as well as, the number of references should be shorter. For example, there are lots of brackets with more than 3 references (one with more than 10). As a recommendation, the most significant references should be included and if all are relevant, their contribution should be pointed out specifically.

Response:

We would like to thank the reviewer for the comment. We would like first to explain the contribution of the paper and then, we comment on the number of references included in our paper.

First, as we already stated in the introduction, “Our study strives to contribute to the debate on the productivity effects of public subsidies through the counterfactual analysis of the effects of the EU public policy”, and its added value is beside the empirical findings in:

1)     In a structured review of the previously published studies showing the previously used empirical approach, productivity variables, findings, period and sample, together with the country of analysis (Table 1)

2)     In combining three different productivity measures and pointing out to the need that the future research should also consider various productivity indicators like total factor productivity (TFP), as already stated in the text.

“In contrast to previous studies that have in view only labour productivity as one of the performance indicators, we solely focus on productivity effects by employing more productivity indicators in one analysis, namely the production efficiency indicator, labour productivity and total factor productivity (TFP), which increases the robustness of our findings. “

Second, the reviewer is concerned about the number of references included in our paper. To be completely honest, we are quite surprised to read such a comment. Citing the existing research is very important. If there are similar studies related to the topic published before, we need to acknowledge them. This is how the research works – we build on what was done in the past, and by very small steps, we push the field forward. By citing the relevant studies that have been published in the past, we also increase the visibility of our study.

Reviewer:

The theoretical framework should include the most significant theories that support the positive/negative effect of public grants on a firm´s productivity, in addition to the discussion of the empirical results from previous research. No hypotheses are included.

Response:

We would like to thank the reviewer for such a comment. We are very again sorry to say that, but the reviewer must have missed the description of theories in our section titled “Theoretical Background”. We have described the reasons for both, positive and negative impacts of public subsidies through the concepts of technological efficiency, economies of scale and wealth effects of subsidies when it comes to the positive impacts of grants, and we have also explained potential reasons for the negative effects of public interventions in light of the allocation inefficiencies and crowding out effects of the subsidies.

Nevertheless, to be even more straightforward, we added to the theoretical section the concept of resource-based view (RBV) of the firm, which assumes that more resources may boost firm´s competitiveness and production efficiency.

In line with the reviewer´s recommendation, we have also included a tested hypothesis into section 3.1 to be even more transparent.

Tested Hypothesis: The firms supported by public grants from the OPEI reported after the end of intervention higher values of productivity indicators, compared to non-supported firms and compared to the period before the public intervention.

Reviewer:

Regarding to the sample, I would like to say that the results could be really bias due to the imbalance in the sample distribution. In order to overcome this point, several researchers create the control group with a similar number of companies with a similar profile (size, subsector, performance...) of the group with public support. More information about the companies included in the sample should be reported. 

Response:

We would like to thank the reviewer for raising this issue. His/her description perfectly matches the principle of counterfactual impact evaluation methodology we use in our paper. It seems that the reviewer must have missed the methodology explained in our paper, and therefore we try to repeat/summarise the most important steps we did.

Very nice guide towards the impact evaluation is a publication by Khandker et al. (2010) from the World Bank, and we use it beside other methodological papers and reports as a justification of our approach. By conducting logistic regression (Table 6), we calculate the propensity score (likelihood of being supported), which is regressed on all available firm-level characteristics we have (as very carefully described in section 4.1), and as we already written in the previous version of our paper

“We control for Year of Registration, Legal Form, Company Size, Region, Sector and pre-intervention financial indicators (Profit/Loss, Total Assets, Trade Margin, Personnel Costs and Debt Ratio).”

By estimating the probability of being supported we match (make twin firms), based on these characteristics mentioned above. The matching quality diagnostics reported in Table 7 show that there are no statistical differences between both groups when it comes to all above-mentioned characteristics. In other words, we managed to match for each of the supported companies a non-supported partner company with similar characteristics. 

Khandker, S. R., Koolwal, G. B., Samad, H. A. Handbook on impact evaluation: quantitative methods and practices; World Bank Publications, 2010. DOI: https://doi.org/10.1596/978-0-8213-8028-4.

Reviewer:

It is surprising that in Table 3 appears the possibility that some companies are large ones (+250 employees). Considering that all the theoretical discussion included in the paper is about SMEs, this fact should be considered.

Response:

We would like to thank the reviewer for mentioning this. This is caused by the initial sample structure described in section 3.2 Sample and Variables. Our paper deals with the effects in one particular industry and thus our population are all firms operating in the industry. According to programme rules, SMEs were mainly supported by the programme. We have not mentioned that specifically and we agree with the reviewer that it is very important to state it clearly. However, to be completely honest, there were a couple (9) of large enterprises supported (exceptions from the programme rules, confirmed by the Ministry of Industry and Trade of the Czech Republic). 95.7% of the supported firms were SMEs. This is what we claim in our former paper, to which we added a reference, for more details, see Dvouletý and Blažková [39].

Dvouletý, O., Blažková, I. Assessing the Microeconomic Effects of Public Subsidies on the Performance of Firms in the Czech Food Processing Industry: A Counterfactual Impact Evaluation. Agribusiness: An International Journal 2018 (in press).

 “The OPEI aimed to mainly support the competitiveness of the small and medium-sized (SMEs) companies.” We have also added the following information into the text “Most of them (95.7%) were SMEs.”

Reviewer:

Regarding to Logistic Regression, there is too many independent variables which means that there may be a collinearity problem. Authors should consider the convenience of summarizing the impact of at least region and subsector variable, that is, finding a point in the literature that help them to divide the sample into less subgroups in both variables. 

Response:

We would like to thank the reviewer for raising the collinearity issue. As we already stated in our text before,

“We keep in our estimates both significant and non-significant independent variables (covariates) to obtain the most precise estimate of the propensity score that will be used for matching subsequently.”

We are aware of the potential threat of the collinearity issue, but we have carefully inspected the level of collinearity between the independent variables based on the results of Variance Inflation Factor (VIF), and we conclude that all values of the test were below the generally accepted threshold of ten. The presented estimates meet the standard econometric assumptions. As for the second issue, we are very keen of the idea of estimating “sub” effects of the programme, and we recommend future scholars to do so in our concluding section. However, as the reviewer know, we have only 157 treated firms which is enough to conduct a reliable econometric analysis for the whole sample, but not much when we would divide the sample for example into the Czech regions (13). To conclude, we agree with the reviewer, but at this point, we cannot do it due to the lack of firms supported by the scheme. 

Reviewer 4 Report

Dear authors,

It was a pleasure to read the paper. It sounds highly scientific, it is difficult to argue with its background and findings.

I'd like to recommend to insert at the conclusion and discussion part more details about your findings. I'd like to read more about your personal opinion about these results. Even in this chapter you point to the results and data  of other scientists. I consider you can add more of your personal opinion, and more overall conclusions based on your collected data. Without (even if is less rigorous) these statements, the paper will contribute to the scientific debate, but will not suggest any improvements to policy makers. Same to many previous researches, your conclusions seems to add just a new file of the growing literature which shows both positive and negative effects of subsidies on productivity.

Author Response

Dear Reviewer,

Thank you very much for the time and efforts you invested in our manuscript, and we hope that we have been able to address all your comments. We have revised our manuscript according to your comments, and we explain the changes in light of comments below. All changes in the manuscript are tracked.

Reviewer´s Comments:

Dear authors,

It was a pleasure to read the paper. It sounds highly scientific, it is difficult to argue with its background and findings.

I'd like to recommend to insert at the conclusion and discussion part more details about your findings. I'd like to read more about your personal opinion about these results. Even in this chapter you point to the results and data  of other scientists. I consider you can add more of your personal opinion, and more overall conclusions based on your collected data. Without (even if is less rigorous) these statements, the paper will contribute to the scientific debate, but will not suggest any improvements to policy makers. Same to many previous researches, your conclusions seems to add just a new file of the growing literature which shows both positive and negative effects of subsidies on productivity. 

Response:

We would like to thank the reviewer for raising this issue. We really aimed to go beyond “coming with another piece of evidence”, and thus, we believe we have very practical research implications for the whole community. These include: 1) Focusing more on the support of firms with a high-growth potential, 2) Boosting rather an entrepreneurial ecosystem instead of supporting a large pool of “average firms”, 3) Using productivity indicators including TFP to assess the impact of public programmes, 4) To compare findings over sub-groups of supported entrepreneurs. We hope that these all practical implications might serve as an important lesson for the community and they also reflect our personal opinion on this issue (although the scientific references wrap it). But is it not the way how we do the research?

Reviewer 5 Report

The paper and study is very well designed and refers to the important topic present among EU-members. The empirical part is strongly backed by theoretical approach, as well as by up-to-date statistical tools. 

Paper can be accepted, as it is, however I would suggest three upgrades (nevertheless I don't expect them to be implemented):

1.     Authors mention prior studies which is extensive in terms of quantity, however it would be valuable to provide more qualitative information on these previous research within the text.

2.     Why Czech food industry is relevant for the study? What is its significance in Czech economy and/or in EU total agriculture output? Some data justifying Authors' focus on this particular industry would be desirable.

3.     Authors adopted 2-year time lag for the analysis of the impact of EU support on firms' productivity. Why two years? Perhaps, additional statistical analysis with different time lag (3 or 4 as currently the only ones possible) might provide additional results and hence significant conclusions.

Author Response

Dear Reviewer,

Thank you very much for the time and efforts you invested in our manuscript, and we hope that we have been able to address all your comments. We have revised our manuscript according to your comments, and we explain the changes in light of comments below. All changes in the manuscript are tracked.

Reviewer´s Comments:

The paper and study is very well designed and refers to the important topic present among EU-members. The empirical part is strongly backed by theoretical approach, as well as by up-to-date statistical tools. 

Paper can be accepted, as it is, however I would suggest three upgrades (nevertheless I don't expect them to be implemented):

1.     Authors mention prior studies which is extensive in terms of quantity, however it would be valuable to provide more qualitative information on these previous research within the text.

Response:

We would like to thank the reviewer for this proposition. Given the space of already quite long manuscript, we have decided to highlight (beside the last three paragraphs in section 2 Theoretical Background) by the review of evidence (Table 1) that most of the previously published studies do not consider more than one or two indicators and that the methodology is not always transparent. We believe that engaged scholars could mine more interesting information from Table 1.

2.     Why Czech food industry is relevant for the study? What is its significance in Czech economy and/or in EU total agriculture output? Some data justifying Authors' focus on this particular industry would be desirable.

Response:

We would like to thank the reviewer for this proposition; we have already tried to explain in the introduction, the specificities of the Czech manufacturing industry, however, in line with the reviewer´s proposition we have added the following sentence:

“Moreover, it is worth mentioning that the share of the food processing industry on the value added of the whole Czech manufacturing industry was 7.5%, and on employment, it was 9.2% in 2016 according to the Czech Statistical Office (c.f. Blažková and Dvouletý [39]).”

3.     Authors adopted 2-year time lag for the analysis of the impact of EU support on firms' productivity. Why two years? Perhaps, additional statistical analysis with different time lag (3 or 4 as currently the only ones possible) might provide additional results and hence significant conclusions.

Response:

We would like to thank the reviewer for raising this issue. We fully agree with his/her opinion and we have already placed this information twice in the section “Discussion and Conclusions”. Concretely, we had the following sentences in the text and thus, we believe that the reviewer has just missed this information in our original text.

“. Finally, given the fact that our results should be interpreted as revealing the short-term effects of subsidies on productivity, one may expect that some of the subsidized firms will go on in the long term to generate larger productivity gains leading to social welfare benefits and positive market and technological spillovers, as supposed by Howell [45]. Therefore, this empirical evaluation should be repeated in a long term and the results of short-term and long-term effects on productivity should be compared.”

“We also believe that, it would be very appropriate to extend the time period of the analysis as more data become available, and to investigate the effects of investment subsidies on the firms' productivity in the long term, since benefits from such innovation-targeting schemes are likely to be expected over longer time frame [35, 116].”

We would be very glad to study long-term effects already in this paper, however, the firm-level data covering the newer years are still not available. 

Resubmission

Round  1

Reviewer 1 Report

Dear Authors.

In the assessment of the paper submitted for the review, I specifically focused on the discussed issues, applied methodology, the substantive content of the paper and its structure.

The subject area discussed in the paper is important and topical but in my opinion is not consistent with the profile of the Journal. There is no connection to Sustainability. Please  explain why and how your study can contribute to a more sustainable society. 

Despite this, the reviewed paper has a lot of values and has a scientific nature. The research procedure adopted by the Authors has complex character. The value of the study results from appropriate  combination of literature studies with the results of an empirical research, which was conducted on a sample of 157 firms. The structure of the paper is clear.

Unfortunately, despite that the bibliography contains lots of papers (117),  very often the authors misuse the references. It is unacceptable that to one very general sentence authors assigned 15 references! (lines 45-50). In my opinion the authors should use no more than 3 references to one sentence. I permit also to suggest to check whether these articles are important.

I am eager to read the new version of the paper, which is both novel and appealing but in present form is not to much connection to profil of the Journal.

Author Response

Dear Reviewer, please, find attached our reply to your review report. 

Reviewer 2 Report

Dear authors,

As a reviewer, I found your research in decent quality and I have no important comment, unless that I recommend you to develop some research questions and then answer to these questions by your findings. in that case, reading of the paper and understanding of its structure would be much easier for the reader. this recommendation is not imperative, and is a suggestion. you can accept and you can decline.

Best of luck!

Author Response

(The authors gave the same response as above.)

Reviewer 3 Report

Please find attached the review report.

Author Response

Dear Reviewer, please, find attached our reply to your review report. 

Round  2

Reviewer 1 Report

Dear Authors.

Thank you for your answer. 

Unfortunately, despite that the bibliography contains lots of papers (117), very often the authors misuse the references. It is unacceptable that to one very general sentence authors assigned 15 references! (lines 51-52). In my opinion the authors should use no more than 3 references to one sentence. 

This should be corrected.

Author Response

Dear Reviewer, please, find attached our reply to your review.

Reviewer 3 Report

The author(s) provided a revised version of the manuscript and explained properly the requested amendments. The paper improved significantly, such as it is worth publishing. Hence, I recommend paper acceptance in present form.

Author Response

Dear Reviewer, please, find attached our reply to your review. 

Round  3

Reviewer 1 Report

Deser Author.

Thank you very much for your response.